# How Employee Job Burnout, Work Engagement, and Turnover Intention Relate to Career Plateau during the Epidemic

Yang Bai [1], Jinquan Zhou [2,*] and Wenjin He [3]

1 College of Management and Economics, Tianjin University, Tianjin 300072, China; 992039@hainanu.edu.cn
2 Center for Gaming and Tourism Studies, Macao Polytechnic University, Macao 999078, China
3 Center of Control and Test Beijing Institute of Technology—Zhuhai Campus, Zhuhai 519088, China; 17234@bitzh.edu.cn
* Correspondence: jqzhou@mpu.edu.mo

**Abstract:** In light of the impact of the COVID-19 epidemic on organizations and company human resource policies, multiple changes have been made to employee work behaviors. This paper developed a concept model on career plateaus, job burnout, work engagement, and turnover intention and examined it through a non-random sampling survey of 285 employees at resorts in Macao. The results revealed that career plateaus positively impact job burnout and turnover intention, and job burnout positively impacts turnover intention. The study found that career plateau negatively impacts work engagement and positively impacts turnover intention, and work engagement negatively influences turnover intention. Job burnout and work engagement partially mediate the relationship between career plateau and turnover intention. Training negatively moderates career plateau toward work engagement, and job rotation moderates career plateau toward turnover intention. Accordingly, organizations should consider the impact on employees' careers when designing training and job rotation policies in response to the epidemic.

**Keywords:** career plateau; work engagement; job burnout; turnover intention; job rotation; training; COVID-19

## 1. Introduction

Job burnout prevalently exists across various industries (Rawolle et al. 2016) and can harm employees and the organizations they work for (Maslach et al. 1997). Among these effects are negative work attitudes (Kim 2016; Tarcan et al. 2017) and physical and mental health problems (Khamisa et al. 2015; Duan-Porter et al. 2018). Even though academics have widely recognized the importance of job burnout, significant differences still require urgent attention (Maslach and Leiter 2008). It is important to note that there is a controversial definition of job burnout in addition to work engagement that remains unclear (Maricuțoiu et al. 2017). Therefore, work engagement and job burnout are inseparable. This relationship affects individuals' work performance, including energy, involvement, and productivity. This debate focuses on whether work engagement is the opposite of job burnout, representing two ends of the same spectrum (Goering et al. 2017). Schaufeli et al. (2002) presented work engagement as a positive and fulfilling psychological state associated with work. It includes feelings of vitality, dedication, and focus, unlike the negative aspects of job burnout. Individuals with low levels of burnout may not necessarily have a high level of engagement in the workplace, and vice versa (Schaufeli and Salanova 2011). Therefore, the two concepts should be considered independent. Subsequently, dedicated measurement tools were developed to assess work engagement, namely the Utrecht Work Engagement Scale (UWES) and the Job Engagement Scale (JES) (Taris et al. 2017). However, regardless of their antecedents and consequences, job burnout and work engagement share similarities and differences. In this regard, it is more meaningful to compare the antecedents and consequences of the two variables, both theoretically and practically.

Furthermore, extensive research encompasses the potential antecedents of burnout and work engagement (Schaufeli and Salanova 2014). Antecedents of job burnout include job demands such as overload, time pressure, and long working hours, in addition to job-related factors such as insufficient social support from colleagues and supervisors, insufficient feedback, and limited participation in decision-making. Antecedents of work engagement include challenging demands such as workload, time urgency, psychological requirements, and responsibility. Researchers studying career plateaus pay more attention to their negative effects (Lapalme et al. 2009). According to Allen et al. (1998), when employees perceive that their performance, contributions, and abilities are no longer appreciated or acknowledged by the organization, they may suffer from dysfunction and psychological distress. This can ultimately lead to inefficiency and ineffectiveness within the organization. For example, studies have found a relatively stable correlation between career plateaus and job burnout (Lemire et al. 1999; Allen et al. 1998). Career plateaus can impact work engagement through various factors (McCleese and Eby 2006; Lentz and Allen 2009; Wang et al. 2014). Therefore, the career plateau may involve common factors that impact burnout and work engagement but have been overlooked by researchers. Therefore, this study examines career plateau as the dependent variable, explores its correlation with burnout and work engagement, and aims to fill this research gap by making comparisons.

Moreover, significant research has been conducted on the effects of burnout and work engagement (Schaufeli and Salanova 2014). The consequences of job burnout involve employee health, including anxiety, depression, and psychosomatic disorders. Additionally, job burnout can lead to poor organizational commitment, increased employee turnover, higher sick leave rates, and decreased job performance. Turnover intention is an outcome variable in job burnout and work engagement that refers to the actual behavior of employees who may choose to stay or quit their job (Allen et al. 2005). Job-related factors, personal and external factors can affect employee turnover (Cotton and Tuttle 1986). Numerous studies have shown that turnover intention can result from job burnout and work engagement. Therefore, this study explores the relationship between career plateau and turnover intention, with a perspective on analyzing work engagement and job burnout as mediating factors. The study aims to investigate the mechanism of action between these variables and compare the relationship between job burnout and work engagement in the context of career plateau and turnover intention to fill this research gap.

In addition, changes in the organizational environment lead to corresponding adjustments in human resource management strategies. The effects of these adjustments can be seen in employees' career plateaus and their attitudes toward the organization and subsequent output. The rapid spread of COVID-19 across the globe (Dryhurst et al. 2020) has made the disease a global health emergency with severe consequences for the worldwide economy (Naseer et al. 2022). COVID-19 has significantly impacted Macau's gambling and tourism industry, causing specified damage to gaming companies' operations. Several casinos have had to close because of the epidemic, and the effects on the gaming and tourism industries may be long-term or permanent. To cope with the adverse impact of the epidemic, many resorts have adopted measures such as salary cuts, training, job rotation, and unpaid leave. This may change employees' career plateaus and attitudes to work, and impact their cognition of job burnout and work engagement. Salary level or job bottlenecks, especially for career plateaus, can lead to a loss of motivation and a severe reduction in work performance, which are detrimental to the organization and its employees (Kwon 2022). Therefore, human resource management intervention that adapts to environmental changes, such as training and job rotation, will impact employee burnout and work engagement and affects the relationship between career plateaus and turnover intentions. It is a novel theoretical and practical requirement for this research.

As mentioned above, this research compares the concepts and theoretical and practical differences between job burnout and work engagement. It also verifies the relationship between career plateau as an antecedent variable and turnover intention as an outcome variable with job burnout and work engagement. It analyzes the impact of training and job

rotation following human resource management in response to the COVID-19 epidemic on career plateaus and organizational behaviors. This study can supplement and perfect employee career management theory and practice.

## 2. Literature Review

### 2.1. Career Plateaus and Turnover Intention

Career plateau refers to a specific stage in an individual's career that involves employee promotion, mobility, and responsibility in an organization (Ference et al. 1977). However, it is not enough to understand career plateaus only from a promotion perspective. Employees move within the organization, and the stagnation of career development includes stagnation in vertical and horizontal mobility (Veiga 1981). Therefore, a career plateau remains a lack of career changes closely related to individual promotion and change in the workplace (Rotondo and Perrewe 2000). In addition, a career plateau can be regarded as the peak point of an individual's career, the relative termination of work responsibilities and challenges in an upward movement, and a stagnation period in the individual's career, but not everyone has to go through a career plateau. Bardwick (1986) proposed two types of career plateau concepts: structure plateau and content plateau. Milliman (1993) proposed a two-dimensional structural division of career plateau: hierarchical and job content plateau, which was widely adopted by follow-up research. Smith-Ruig (2009) believed that career plateaus include three dimensions: objective plateaus, internal subjective plateaus, and external subjective plateaus. The internal subjective plateau refers to employees' perceptions of advancement opportunities within an organization. The external subjective plateau relates to employees' perceptions of whether they can succeed in an organization or find better job opportunities outside of the organization.

Turnover intention is the psychological motivation of employees who want to find other job opportunities but have not left the organization (Allen et al. 2005), which is linked to three factors: work, personal, and external (Cotton and Tuttle 1986). Job-related factors involve salary, performance, job satisfaction, and organizational commitment, among which job satisfaction and organizational commitment have received the most attention because they directly and strongly affect turnover intentions (Futrell and Parasuraman 1984), as well as gender, marital status, age, education level, monthly salary, position, etc. Among these, gender and marriage have received the most attention. External-related factors include economic development level, labor market conditions, competition in the same industry, the employment system, industry bias, etc. (Zhang 2016).

When employees cannot improve their skills or knowledge of their fields, they will feel less hope for their careers and leave their jobs. When employees know they have hardly any room for promotion, they may quit the company to find a better development environment (Godshalk and Fender 2015). Tremblay et al. (1995) found that turnover intentions increased with higher hierarchical plateaus. Similar findings were reported in studies by Lemire et al. (1999) and Xie et al. (2016). In addition, Lentz and Allen (2009), Wang et al. (2014), and Drucker-Godard et al. (2015) all reported a positive relationship between career plateaus and turnover intention. Therefore, the following hypotheses are proposed:

**H1:** *Career plateau has a positive effect on turnover intention.*

### 2.2. Career Plateau, Job Burnout, and Turnover Intention

Job burnout is a state of work failure, energy exhaustion, and physical and mental exhaustion caused by work that requires excessive ability, energy, and resources from an individual. It is a series of symptoms related to work. Freudenberger (1974) suggests that an employee's excessive occupational fatigue is manifested as an undesirable form of work attitudes and behaviors, and it will continue to deepen the process of Cherniss and Cherniss (1980). Leiter and Maslach (1988) believed that job burnout is a symptom of three subdivided dimensions: emotional exhaustion, low personal accomplishment, and a lack of human touch. Maslach et al. (2001) believed that job burnout subdivision dimensions are negative snub, emotional depletion, and low career performance. Burnout remains a

persistent, negative psychological state that frequently occurs in ordinary people and is closely related to occupations. Exhaustion is one of its most significant characteristics, the same occupation that causes decreased motivation, effectiveness, restlessness, and negative attitudes. It is also associated with other undesirable phenomena (Schaufeli et al. 2002). Lee and Ashforth (1993) believed that job burnout resulted from excessive demands on professional service employees' resources and capabilities, which made employees unable to better deal with service objects. Employees were chronically tired and felt helpless. Meaningful activities in the workplace are affected by negative attitudes and beliefs from burnout (Gray-Stanley and Muramatsu 2011). If the enterprise members' ideals, ambitions, and high hopes are not fulfilled, it will lead to job burnout (González-Morales et al. 2012).

Furthermore, present academic research has focused on the relationship between career plateaus and job burnout (Salvagioni et al. 2017). Feldman and Weitz (1988) proposed that job burnout leads to a career plateau, and job performance serves a mediating role in the middle. They believe career plateaus are caused by job burnout. On the contrary, Wolf (1983) argued that a career plateau affects employees' aggressiveness and work enthusiasm, which then makes employees feel job burnout; that is, a career plateau leads to job burnout. Burke (1989) believed that career plateaus could relate to job burnout. Long working hours and high work intensity are crucial reasons for career plateaus. When the company does not reward employees with corresponding promotions or salary increases, employees will feel burnout, and the career plateau will further increase and deepen (Edú-Valsania et al. 2022). Job burnout affects employees' participation in work, and their enthusiasm for work will also decrease, resulting in a decline in job satisfaction, weakening personal commitment to the organization, and even forming personal turnover intentions (Hofstetter and Cohen 2014). Beheshtifar (2017) investigated the relationship between hospital staff's career plateau and job burnout and found that nurses' career plateau (two dimensions of the content plateau and hierarchical plateau) has a significant relationship with job burnout. Fayyazi and Ziaei (2015) found through a sample study of 537 university librarians that librarians' career plateaus (hierarchical plateaus and content plateaus) and job burnout rates were higher than the average. Therefore, career plateau may significantly impact job burnout, and the higher the subjective job plateau, the higher the likelihood of job burnout (Tremblay et al. 1995; Xie and Long 2008). Therefore, the following hypotheses are proposed:

**H2:** *Career plateau positively impacts job burnout.*

**H3:** *Job burnout positively impacts turnover intention.*

**H4:** *Job burnout mediates the relationship between career plateau and turnover intention.*

### 2.3. Career Plateau, Work Engagement, and Turnover Intention

Work engagement was proposed as a concept recognized by practitioners and researchers (Schaufeli et al. 2002), who believed that work engagement is a positive and complete emotional and cognitive state related to work, including work vitality, work dedication, and work focus (Roberts and Davenport 2002). Work vitality is similar to work motivation, while work dedication is more akin to work engagement (Mauno et al. 2007). Work engagement is a motivational construct that makes psychological sense, and self-engagement in job roles only makes sense if people feel like they are contributing to work and getting something out of it (Antony 2018). Saks and Gruman (2011) developed a model of work engagement that regards increased work engagement as a precursor to improving job performance. Studies have shown that career plateaus negatively impact job performance and work engagement (Huaman-Ramirez and Lahlouh 2022). Career plateaus negatively affect work engagement and other behavior outcomes (Nachbagauer and Riedl 2002; Yang et al. 2019). Therefore, a career plateau negatively impacts work engagement, which may be affected by other influencing factors (McCleese and Eby 2006; Lentz and Allen 2009; Wang et al. 2014). Therefore, the following hypotheses are proposed:

**H5:** *Employee career plateau negatively impacts work engagement*

**H6:** *Work engagement negatively impacts turnover intention.*

**H7:** *Work engagement mediates the relationship between career plateau and turnover intention.*

*2.4. Training, Job Rotation, Career Plateau, Work Engagement, and Job Burnout*

Organizational support is a crucial factor affecting employees' psychological cognition, which can bring significant psychological resources to employees and affect their career development (Kraimer et al. 2011). Organizational support concerns whether the organization offers stagnant employees anything to compensate them for promotions or job challenges (Abele et al. 2011). A lack of support can explain why career-stagnant workers conduct poor work outcomes (Karatepe and Olugbade 2016). Organizational support mediates the negative relationship between hierarchical plateaus and job performance and satisfaction (Lin and Chen 2021). Armstrong-Stassen and Ursel (2009) provided that organizational support mediates career plateaus toward job contentment.

Organizations support offering employees training and job rotation rather than promotion while coping with career plateau issues (Rotondo and Perrewe 2000). Organizations should buffer the relationship between perceived career plateaus and a lack of organizational support and ultimately reduce the negative impacts of career plateaus (Kao et al. 2022). Therefore, training activities are pivotal in encouraging employees' learning and development and, to a certain extent, enhancing employees' work engagement (Tripathy 2020). In addition, the organization supports job rotation within the organization. Employees with stagnant career development can either perform vertical mobility and get promoted to higher levels or perform horizontal mobility to learn new knowledge and skills in existing jobs and improve their work capabilities, laying the foundation for future career development, reducing career plateaus, and increasing turnover intentions (Campion et al. 1994; Ference et al. 1977). Job rotation may be a temporary measure during COVID-19 (Kwon 2022). Therefore, the following hypotheses are proposed:

**H8a:** *Training moderates career plateau toward job burnout.*

**H8b:** *Training moderates career plateau toward work engagement.*

**H9a:** *Job rotation moderates career plateau toward job burnout.*

**H9b:** *Job rotation moderates career plateau toward work engagement.*

The research model was formed in Figure 1.

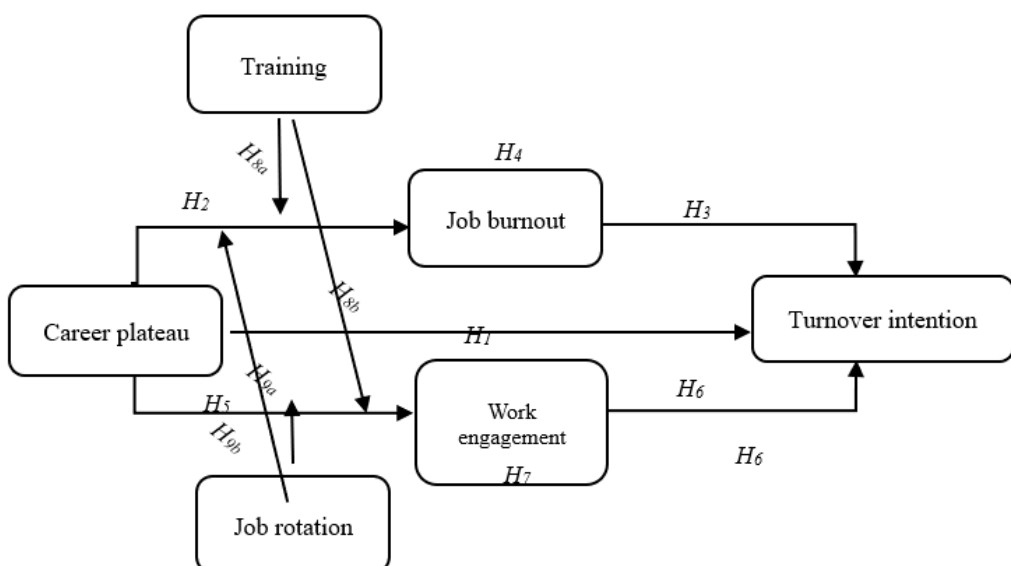

**Figure 1.** Conceptual framework of the proposed moderation mediation model.

## 3. Materials and Methods

### 3.1. Scale

The career plateau measurement mainly refers to and draws on Milliman's (1993) career plateau scales, including hierarchical plateau (2 scales) and content plateau (3 scales). The work engagement measurement adopts the scale of Schaufeli et al. (2002), including work vitality (3 items), work concentration (3 items), and dedication (3 items). Job burnout measurement refers to the scale of Maslach et al. (1997), which contains 7 scales. The turnover intention measurement t refers to the scale of Singh and Loncar (2010), which has four scales. Training measurement is on one scale: "I received training during the COVID-19 epidemic ", and job rotation measurement is on one scale: "I received job rotation during the COVID-19 epidemic". The measurement instrument was first developed in English and translated into Chinese by two bilingual professionals proficient in both languages. The revised questionnaire was created using a 5-point scale for the final survey.

### 3.2. Data

The survey team conducted a questionnaire survey with the help of employees from six resorts in Macao. The survey was conducted at the employees' exit gates of six resorts from 8 January 2022 to 2 February 2022. A total of 350 questionnaires were distributed, and 285 valid questionnaires were recovered, for an effective rate of 81.4%.

### 3.3. Sample

Table 1 presents the sample's general characteristics. The absolute value of the bias of all measurement items was less than 3, and the absolute value of the kurtosis was less than 10. Thus, it can be considered that each item of the sample data meets the normal distribution requirements, and further SEM analysis can be carried out (Gao et al. 2008).

**Table 1.** The sample description (*n* = 285).

|  |  | Frequency | Percentage |
|---|---|---|---|
| Gender | Female | 141 | 49.47 |
|  | Male | 144 | 50.53 |
| Marriage | Unmarried | 63 | 22.1 |
|  | Married | 222 | 77.9 |
| Age | 21–30 | 66 | 23.2 |
|  | 31–40 | 130 | 45.6 |
|  | 41–50 | 64 | 22.5 |
|  | 51 and above | 25 | 8.8 |
| Education | Junior high school | 107 | 37.5 |
|  | High school | 111 | 38.9 |
|  | University and above | 67 | 23.5 |
| Position | Dealer | 208 | 73 |
|  | Supervisor | 65 | 22.8 |
|  | Pit manager | 12 | 4.2 |
| Seniority | 4 years and below | 83 | 29.1 |
|  | 5–10 years | 149 | 52.3 |
|  | 11–15 years | 49 | 17.2 |
|  | 15 years and above | 4 | 1.5 |
| Firm | Macau Entertainment | 41 | 14.4 |
|  | MGM China | 61 | 21.4 |
|  | Melco Entertainment | 56 | 19.6 |
|  | Wynn China | 71 | 24.9 |
|  | Sands China | 41 | 14.4 |
|  | Galaxy Entertainment | 15 | 5.3 |

*3.4. Measures*

Cronbach's alpha measures the internal reliability of the questionnaire. Cronbach's alpha was 0.777 for career plateau, 0.831 for work engagement, 0.823 for job burnout, and 0.854 for turnover intention. The coefficient of Cronbach's alpha for each latent variable was above 0.7, indicating the questionnaire's high reliability and internal consistency. The overall variance explained in the questionnaire's common method variance (CMV) test was 16.791% and less than 50%. There was no issue with common method bias.

Table 2 shows that all latent variables have a combined reliability (CR) value higher than the critical value of 0.7, indicating the high reliability of the measurement model after analysis. The model's validity was tested through a convergent and discriminant validity test. The factor loading values for each observation index ranged from 0.425 to 0.832, with *p*-values below 0.001. The average extraction variance of each latent variable ranges from 0.360 to 0.591 (Fornell and Larcker 1981; Lam 2012). This shows that the measurement model had strong convergence validity.

**Table 2.** Confirmatory factor analysis.

|  | Item | Loading | SMC | CR | AVE |
|---|---|---|---|---|---|
| Career plateau | I have limited space for further promotion at the company. | 0.824 | 0.679 | 0.775 | 0.423 |
|  | I could not have gotten a higher position in this company. | 0.832 | 0.692 |  |  |
|  | My current job in the company can use my talents. | 0.455 | 0.207 |  |  |
|  | I will accept any work arrangement to stay in the company. | 0.520 | 0.270 |  |  |
|  | I am happy that I chose to work for the company. | 0.517 | 0.267 |  |  |
| Turnover intention | I will leave my current company if given the opportunity. | 0.816 | 0.666 | 0.852 | 0.591 |
|  | I want to work for another company. | 0.757 | 0.573 |  |  |
|  | I have to consider leaving your current company. | 0.723 | 0.523 |  |  |
|  | I often have the idea of leaving your current unit. | 0.775 | 0.601 |  |  |
| Work engagement | When I work, I feel strong and energized. | 0.592 | 0.350 | 0.833 | 0.360 |
|  | When I wake up in the morning, I want to go to work. | 0.654 | 0.428 |  |  |
|  | I can work long hours at a time. | 0.575 | 0.331 |  |  |
|  | When I'm working, I forget about everything around me. | 0.508 | 0.258 |  |  |
|  | When work is stressful, I feel happy. | 0.592 | 0.350 |  |  |
|  | I am immersed in my work. | 0.565 | 0.319 |  |  |
|  | I feel that what I do is purposeful and meaningful. | 0.498 | 0.248 |  |  |
|  | My work inspires me. | 0.718 | 0.516 |  |  |
|  | I am proud of the work I do. | 0.658 | 0.433 |  |  |
| Job burnout | I feel a lack of rest throughout the year. | 0.726 | 0.527 | 0.825 | 0.404 |
|  | My work has no autonomy and requires the supervisor to check in. | 0.607 | 0.368 |  |  |
|  | I did more work but received not much. | 0.722 | 0.521 |  |  |
|  | Individuals in the company do not take care of each other. | 0.626 | 0.392 |  |  |
|  | I think my company makes money first, profit-oriented. | 0.592 | 0.350 |  |  |
|  | I am inconsistent with the company's management philosophy. | 0.568 | 0.323 |  |  |
|  | I feel like the company is not fair. | 0.591 | 0.348 |  |  |

Table 3 shows that the square root of AVE had a higher value than the construct's correlation with other constructs, indicating strong convergent validity (Hair et al. 2014). This supports the discriminant validity measurement.

Finally, the value of model fit reported an $X^2/df$ of 1.954 (527.553/270), a GFI of 0.871, an AGFI of 0.845, an NFI of 0.814, a TLI of 0.887, a CFI of 0.899, and an RMSEA of 0.058. The confirmatory factor analysis in this paper indicates an acceptable overall fit of the model, meeting all standard indicators.

**Table 3.** Mean values, stand deviation, and correlation coefficient.

| N = 285 | MD | S.E | Correlations | | | | | | | | |
|---|---|---|---|---|---|---|---|---|---|---|---|
| | | | 1 | 2 | 3 | 4 | 5 | 6 | 7 | 8 | 9 |
| **CP** | **3.106** | 0.823 | **(0.65)** | | | | | | | | |
| JB | 3.192 | 0.727 | 0.504 ** | **(0.60)** | | | | | | | |
| WE | 2.945 | 0.742 | −0.258 ** | −0.205 ** | **(0.636)** | | | | | | |
| TI | 2.950 | 0.928 | 0.335 ** | 0.287 ** | −0.652 ** | **(0.769)** | | | | | |
| Age | 2.170 | 0.884 | −0.136 * | −0.011 | 0.196 ** | −0.237 ** | 1 | | | | |
| Gender | 0.510 | 0.501 | −0.079 | −0.044 | −0.017 | 0.055 | −0.066 | 1 | | | |
| Marriage | 0.780 | 0.416 | −0.012 | −0.102 | −0.032 | 0.01 | −0.167 ** | −0.003 | 1 | | |
| Education | 1.860 | 0.767 | 0.05 | 0.038 | −0.024 | 0.1 | −0.265 ** | −0.085 | −0.012 | 1 | |
| Position | 1.310 | 0.548 | −0.181 ** | −0.082 | 0.084 | −0.089 | 0.015 | 0.013 | 0.041 | 0.267 ** | 1 |
| Seniority | 2.310 | 1.088 | −0.626 ** | −0.344 ** | 0.146 * | −0.288 ** | 0.290 ** | 0.039 | −0.045 | −0.036 | 0.235 ** |

* $p < 0.05$; ** $p < 0.001$; CP = Career plateau; JB = Job burnout; WE = Work engagement; TI = Turnover intention; The square root of AVE (bold values).

## 4. Results

Table 4 shows the results of the hierarchy of career plateau, work engagement, and job burnout on turnover intention. When seniority was added to the model, the research showed a relationship between work engagement and turnover intention ($\Delta R2 = 0.38$, $\Delta F = 200.021$), and between job burnout and turnover intention ($\Delta R2 = 0.04$, $\Delta F = 12.851$), indicating seniority had a control effect. The results demonstrate that there is no multi-collinearity among the independent variables. Career plateau positively affected turnover intention (B = 0.203, $p < 0.001$), career plateau negatively affected work engagement (B = −0.642, $p < 0.001$), and work engagement negatively affected turnover intention (B = −0.642, $p < 0.001$). Job burnout positively affected turnover intention (B = 0.213, $p < 0.001$), and career plateau positively affected job burnout (b = 0.154, $p < 0.001$), indicating H1, H2, H3, H5, and H6 were supported.

**Table 4.** The hierarchy regression of career plateau, work engagement, and job burnout on turnover intention.

| | M1 | | | M2 | | | M3 | | |
|---|---|---|---|---|---|---|---|---|---|
| | B | SE | t | B | SE | t | B | SE | t |
| Seniority | −0.288 | 0.049 | −5.064 ** | −0.192 | 0.037 | −4.468 ** | −0.141 | 0.047 | −2.592 * |
| Work engagement | | | | −0.642 | 0.057 | −14.143 ** | −0.628 | 0.058 | −14.256 ** |
| Career plateau | | | | | | | 0.095 | 0.084 | 1.509 |
| ΔR2 | | 0.083 | | | 0.403 | | | 0.004 | |
| ΔF | | 25.646 ** | | | 221.503 ** | | | 2.278 | |
| Seniority | −0.288 | 0.049 | −5.064 ** | −0.215 | 0.051 | −3.619 ** | −0.122 | 0.06 | −1.726 |
| Job burnout | | | | 0.213 | 0.076 | 3.585 ** | 0.154 | 0.082 | 2.398 * |
| Career plateau | | | | | | | 0.203 | 0.18 | 2.336 * |
| ΔR2 | | 0.083 | | | 0.04 | | | 0.017 | |
| ΔF | | 25.646 ** | | | 12.851 ** | | | 5.459 * | |

* $p < 0.05$; ** $p < 0.01$.

Furthermore, SPSS26.PROCESE2.16 software was used to analyze the mediating effect between job burnout, work engagement in the career plateau, and turnover intention. Bootstrapping was used to estimate total, direct, and indirect effects accordingly (see Table 5). The range of the estimated total effect confidence interval, the standardized indirect effect confidence interval, and the standardized direct effect confidence interval did not include 0, which means that the total effect, indirect effect, and direct effect all exist significantly and are positive. Thus, H4 and H7 were supported.

**Table 5.** Mediating effect test.

| Path | Effect | B | SE | 95.0% Confidence Interval | |
|---|---|---|---|---|---|
| | | | | LL | UL |
| CP-WE-MI | Total effect | 0.377 | 0.063 | 0.253 | 0.501 |
| | Direct effect | 0.201 | 0.051 | 0.010 | 0.302 |
| | Indirect effect | 0.176 | 0.041 | 0.099 | 0.260 |
| CP-JB-MI | Total effect | 0.377 | 0.063 | 0.253 | 0.501 |
| | Direct effect | 0.287 | 0.072 | 0.144 | 0.429 |
| | Indirect effect | 0.09 | 0.04 | 0.015 | 0.173 |

CP = Career plateau; JB = Job burnout; WE = Work engagement; TI = Turnover intention.

Finally, SPSS 26. PROCESE 2.16 software was used to analyze the moderating effects of training and job rotation on career plateaus, work engagement, job burnout, and turnover intention (see Table 6). Training negatively moderated career plateau toward job burnout (R = −0.076), and job rotation negatively moderated career plateau toward work engagement (R = −0.103). Therefore, we assumed that H8a and H9b were supported, but H8b and H9a were not supported. Figures 2 and 3 show the moderating effect.

**Table 6.** Examination of the moderating effect of training and job rotation.

| Model 1 | R | R-sq | MSE | F | df1 | df2 | p |
|---|---|---|---|---|---|---|---|
| | 0.516 | 0.267 | 0.392 | 34.052 | 3 | 281 | 0 |
| | coeff | SE | t | p | LLCI | ULCI | |
| constant | 3.193 | 0.037 | 86.055 | 0 | 3.12 | 3.266 | |
| Train | −0.018 | 0.032 | −0.575 | 0.566 | −0.081 | 0.044 | |
| CP | 0.445 | 0.045 | 9.867 | 0 | 0.357 | 0.534 | |
| int_1 | −0.076 | 0.038 | −2.035 | 0.043 | −0.150 | −0.002 | |
| Product terms key: int_1 CP X Train | | | | | | | |
| R-square increase due to interaction(s): | | | | | | | |
| | R2-chng | F | df1 | df2 | p | | |
| int_1 | 0.011 | 4.14 | 1 | 281 | 0.043 | | |
| **Model 2** | R | R-sq | MSE | F | df1 | df2 | p |
| | 0.303 | 0.092 | 0.505 | 9.482 | 3 | 281 | 0 |
| | coeff | SE | t | p | LLCI | ULCI | |
| constant | 2.941 | 0.042 | 69.815 | 0 | 2.858 | 3.024 | |
| JR | 0.022 | 0.032 | 0.678 | 0.498 | −0.041 | 0.084 | |
| CP | −0.226 | 0.051 | −4.41 | 0 | −0.327 | −0.125 | |
| int_1 | −0.103 | 0.038 | −2.697 | 0.007 | −0.177 | −0.028 | |
| Product terms key: int_1 CP X JR | | | | | | | |
| R-square increase due to interaction(s): | | | | | | | |
| | R2-chng | F | df1 | df2 | p | | |
| int_1 | 0.024 | 7.272 | 1 | 281 | 0.007 | | |

CP = Career plateau; JB = Job burnout; WE = Work engagement; TI = Turnover intention.

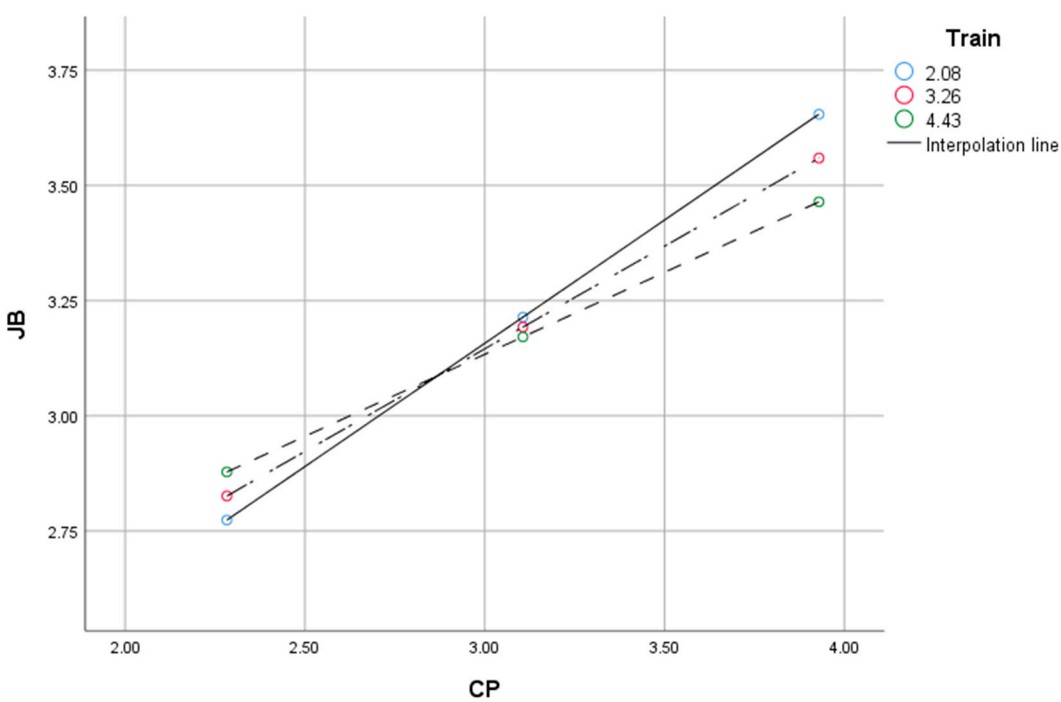

**Figure 2.** The moderating effect of training.

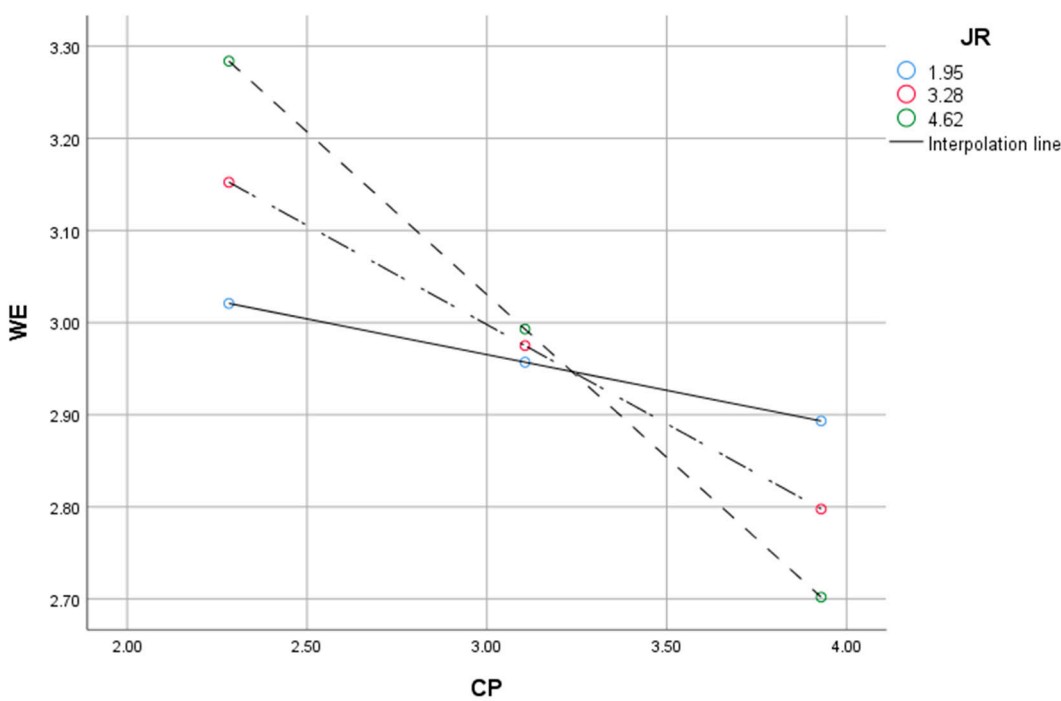

**Figure 3.** The moderating effect of job rotation.

## 5. Discussion and Implications

This research investigates the current literature and presents a theoretical framework for job burnout, work engagement, career plateaus, and turnover intentions. It identifies the differences between job burnout and work engagement in their antecedents and consequences. This research investigates a sample data set of employees in Macau to support the proposed research hypothesis. According to the adjustment of human resources policy under the COVID-19 epidemic, the moderating effects of training and job rotation on career

plateau and job burnout, work engagement, and turnover intention were verified. The detailed analysis and conclusions are as follows.

*5.1. Discussion*

First, the results found that career plateaus affect turnover intentions. They show that individuals show high turnover intentions when at a career plateau. This is consistent with the conclusion of Tremblay et al. (1995), which found that turnover intentions were higher as the level of career plateau increased. Similar findings were reported in studies by Lemire et al. (1999) and Xie et al. (2016). This study shows a significant positive relationship between career plateau and turnover intention and that other influencing factors may affect turnover intention (Lentz and Allen 2009; Wang et al. 2014; Drucker-Godard et al. 2015). This may be because employees at career plateaus have less space for promotion, lack lateral mobility, and lack experience in fulfillment and career development, so they have turnover intention. Therefore, resorts should consider taking available human resources measures like training and job rotation to avoid premature career plateaus for dealers and the resulting turnover.

Second, the results reported that career plateau positively impacts job burnout. Studies have shown that individuals exhibit higher levels of burnout when they feel they are at a career plateau, which is consistent with many previous studies (Edú-Valsania et al. 2022; Beheshtifar 2017; Fayyazi and Ziaei 2015). Career plateau significantly impacts job burnout, and the higher the subjective career plateau, the more severe the job burnout. The results found that job burnout positively impacts turnover intention, meaning that job burnout affects employees' participation in work, and their enthusiasm for work will also decrease, resulting in reduced job satisfaction and weakened personal commitment to the organization, leading to final turnover intention (Tremblay et al. 1995; Xie and Long 2008). Thus, job burnout can be an effective indicator to predict turnover intention. Therefore, resorts should consider taking valid human resources measures to reduce career plateaus' impact on job burnout for dealers' turnover.

Third, the results revealed that career plateaus negatively impact work engagement. This means employees offer low work engagement while staying at a career plateau. This is consistent with many previous research results (Saks and Gruman 2011; Huaman-Ramirez and Lahlouh 2022). Thus, a significant negative relationship between career plateaus and work engagement may affect work engagement through other influencing factors. This may be because employees at career plateaus have less space for promotion and cannot absorb updated knowledge and skills in their current work. They cannot improve themselves, and the organization does not give employees more rights, responsibilities, or organizational resources. The results found that work engagement negatively impacts turnover intention, meaning that high level turnover intention will decrease work engagement. which is consistent with many previous research results (Nachbagauer and Riedl 2002; Lentz and Allen 2009; McCleese and Eby 2006; Wang et al. 2014). Therefore, in such an organizational atmosphere, employees experience such a sense of lack of accomplishment and lack of career development that it is difficult to maintain their loyalty to the organization.

Fourth, the results reported that job burnout and work engagement partially mediate career plateau toward turnover intention. Studies have shown that when individuals are at a career plateau, the impact of employee burnout and work engagement on subsequent turnover intentions is consistent with many previous studies (Xie and Long 2008), indicating that both can be used as predictors. Comparing the direct effects of the two, it shows that the impact of job burnout on turnover intention is 0.201 (0.051), significantly smaller than the impact of work engagement on turnover intention at 0.287 (0.072). Comparing the indirect effects for employees at career plateaus, the impact of job burnout on turnover is 0.176 (0.041), significantly higher than the effects of work engagement at 0.09 (0.04). Thus, job burnout is a better explanation than work engagement for the relationship between career plateaus and turnover intentions. Therefore, researchers can refer to this result when selecting measurement indicators in the models in future research.

Fifth, the study found that training moderates career plateaus and job burnout, and its impact coefficient on job burnout is negative, consistent with many previous studies (Nachbagauer and Riedl 2002; Yang et al. 2019). Under the influence of COVID-19, organizations can effectively reduce the impact of career plateaus on job burnout by organizing training as an organizational support activity (Rawashdeh and Tamimi 2020; Tripathy 2020), thereby reducing employees' turnover intention. Additionally, job rotation moderates career plateaus toward work engagement. The impact coefficient on work engagement is also positive, indicating that appropriate job rotation can reduce the negative impact of career plateau and minimize turnover intention (Lin and Chen 2021). Meanwhile, research shows that training has no significant moderating effect on work engagement, and job rotation has no significant moderating effect on job burnout. Therefore, under the influence of the COVID-19 epidemic, job changes can increase employee engagement, and training can reduce the perception of job burnout.

Finally, the results found that seniority has a significant control effect in this study. Seniority has a controlling impact on career plateau and job burnout and is negatively correlated with job burnout. Seniority has a control effect on career plateau and work engagement and is positively related to work engagement. It shows that senior employees in an organization have low burnout but high work engagement. This may be due to the impact of COVID-19 and the influence of the unique industry context of gambling. This is because the promotion and salary mechanisms of the gaming industry are highly related to seniority. In addition, due to the long-term accumulation of service technology and customer relationships in dealers' work, employees with long experience can achieve positive results without more investment.

*5.2. Implications*

Theoretically, this study examines the differences in the literature and empirical impacts of job burnout and work engagement, their impact on turnover intentions, and their moderating effects between career plateaus and turnover intentions. Especially for service industry employees, it provides strong support for related theories of organizational behavior. In particular, this study finds that job burnout is a better explanation than work engagement for the relationship between career plateaus and turnover intentions. This provides strong evidence for the original theoretical argument in this paper.

In practice, the results of the sample data show that enhancing employees' sense of belonging to the organization is significant for reducing the plateau period of careers under the circumstances of COVID-19, especially for the loss of confidence in job prospects, a lack of passion for work content and results, and a significant reduction in work efficiency (Stoyanova and Iliev 2017; Scott and Leadership 2017). When resorts arrange the development of employee careers, in the case of limited positions or promotion space, it is of practical significance to adopt appropriate human resource strategies to reduce their turnover intention.

The data show that most employees are 35–42 years old. When the job skills become more and more proficient, repetitive work content day after day makes employees feel drained, and the appreciation space is low. They are therefore more likely to reach the career plateau. Even though the employee's work performance and ability are outstanding, they are not recognized due to a lack of breakthroughs. A dealer's salary is higher than the local average in Macau. Although employees will not be dissatisfied with their jobs, a career plateau will positively impact their turnover intentions. Reducing this impact by increasing positive organizational support activities, such as training, may be an effective way for Macau resorts to cope with the external environment in the long run.

*5.3. Limitations*

This paper has several research limitations. First, according to research samples, this paper selected dealers who are the most typical representative of the Macau resorts as the research objects. However, there are certain deficiencies in the number of samples and

the sample commonality. Second, the gaming companies involved in this article include Chinese and foreign capital. There are noticeable differences in corporate culture and management methods. Therefore, it is more meaningful to compare the gaps between the two groups of companies in terms of career plateaus and organizational human resource policies. Third, the epidemic's impact is diverse, including changes in employees' working conditions and income, which may impact career plateau industries. Future models will analyze these factors with more theoretical and practical significance. Fourth, this paper only examines the impact of employees' career plateaus and human resource policies on internal factors. Research may focus on how the corresponding changes in opportunities from outside the organization impact career plateaus in the future, especially the changes in external opportunities caused by environmental factors.

**Author Contributions:** Conceptualization, J.Z. and Y.B.; methodology, J.Z. and W.H.; formal analysis, W.H.; investigation, J.Z.; writing—original draft preparation, Y.B.; writing—review and editing, J.Z. and Y.B.; supervision, J.Z.; project administration, J.Z.; funding acquisition, J.Z. All authors have read and agreed to the published version of the manuscript.

**Funding:** This research was funded by the Macao Polytechnic University Foundation: RP/CJT-03/2021.

**Institutional Review Board Statement:** Not applicable.

**Informed Consent Statement:** Informed consent was obtained from all subjects involved in the study.

**Data Availability Statement:** The research data can be available for request from the Corresponding writer.

**Conflicts of Interest:** The authors declare no conflict of interest.

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
