# Peer review of "How Employee Job Burnout, Work Engagement, and Turnover Intention Relate to Career Plateau during the Epidemic"

_socsci, doi:10.3390/socsci12070394_

Round 1

Reviewer 1 Report

Thanks for the opportunity to review this paper. The paper in general is clearly presented. However, there are some issues need to be clarified. 

1. I am not convinced why the paper starts with the comparison of burnout and job engagement and I don't see the empirical contribution of this discussion. 

2. The previous verified correlation between career plateaus and burnout can be interpreted in different ways. As the previous literature discussed, burnout itself can cause turnover intention, and more significant than career plateaus, I don't see the significance to put burnout as a mediating variable to text the correlation between career plateaus and turnover intention. 

3. How did the Covid environment make a difference?

4. I don't see the rationale of sampling. Why you collect data from those resort exits? How did the data collection method impact the data quality?

5. It will be helpful if the author(s) can present a figure illustrating the theoretical model. 

6. Training as a moderating variable is roughly designed. I believe different kind of training makes a big difference. 

7. Not sure if the 3 item instrument of career plateaus is able to capture objective plateaus, internal subjective plateaus, external subjective plateaus, how?

The paper is presented in fluent english, just minor edits is needed. 

Author Response

Dear Editor, Reviewers,

We appreciate you and the reviewer for your precious time in reviewing our paper and

providing valuable comments. We have carefully considered the comments and tried our best to address every one of them. We hope the manuscript after careful revisions meet your high standards. We welcome further constructive comments if any. Below we provide the point-by-point responses. All modifications in the manuscript have been highlighted in yellow color.

Sincerely,

Yours

Reviewer 2 Report

This is a well done paper and study. I am recommending it for minor revisions based on the following: 

-further contextualize this within the professional niche of the case study because it has important implications. 

-similarly, consider cultural implications of this study and the potential transferability of your findings in other industries and other cultures, as well as across generations since you do mention most employees were between 35-42. 

-the results could be revised to include a few sentences for each hypothesis that are more explicit about the findings and simplify the stats a bit.

-in the discussion section, consider interpreting the findings more deeply rather than just presenting them and digging into what organizations can do with this information and how it can be applied.

Author Response

(The authors gave the same response as above.)
